# Chromosomal neighbourhoods allow identification of organ specific changes in gene expression

**Rishi Das Roy**[1]*, **Outi Hallikas**[1], **Mona M. Christensen**[1], **Elodie Renvoisé**[1], **Jukka Jernvall**[1,2]*

**1** Institute of Biotechnology, University of Helsinki, Helsinki, Finland, **2** Department of Geosciences and Geography, University of Helsinki, Helsinki, Finland

* rishi.dasroy@helsinki.fi (RDR); jernvall@fastmail.fm (JJ)

## Abstract

Although most genes share their chromosomal neighbourhood with other genes, distribution of genes has not been explored in the context of individual organ development; the common focus of developmental biology studies. Because developmental processes are often associated with initially subtle changes in gene expression, here we explored whether neighbouring genes are informative in the identification of differentially expressed genes. First, we quantified the chromosomal neighbourhood patterns of genes having related functional roles in the mammalian genome. Although the majority of protein coding genes have at least five neighbours within 1 Mb window around each gene, very few of these neighbours regulate development of the same organ. Analyses of transcriptomes of developing mouse molar teeth revealed that whereas expression of genes regulating tooth development changes, their neighbouring genes show no marked changes, irrespective of their level of expression. Finally, we test whether inclusion of gene neighbourhood in the analyses of differential expression could provide additional benefits. For the analyses, we developed an algorithm, called DELocal that identifies differentially expressed genes by comparing their expression changes to changes in adjacent genes in their chromosomal regions. Our results show that DELocal removes detection bias towards large changes in expression, thereby allowing identification of even subtle changes in development. Future studies, including the detection of differential expression, may benefit from, and further characterize the significance of gene-gene neighbour relationships.

## Author summary

Development of organs is typically associated with small and hard to detect changes in gene expression. Here we examined how often genes regulating specific organs are neighbours to each other in the genome, and whether this gene neighbourhood helps in the detection of changes in gene expression. We found that genes regulating individual organ development are very rarely close to each other in the mouse and human genomes. We built an algorithm, called DELocal, to detect changes in gene expression that incorporates

**Data Availability Statement:** The DELocal algorithm is freely available here https://github.com/dasroy/DELocal as an R package. The microarray and RNAseq data are deposited in the

NCBI Gene Expression Omnibus with accession codes GSE141907 and GSE142199.

**Funding:** ER and JJ were funded by the Academy of Finland. JJ was funded by Jane and Aatos Erkko Foundation, John Templeton Foundation, and Sigrid Jusélius Foundation. The funders had no role in study design, data collection and analysis, decision to publish, or preparation of the manuscript.

**Competing interests:** The authors have declared that no competing interests exist.

information about neighbouring genes. Using transcriptomes of developing mouse molar teeth containing gene expression profiles of thousands of genes, we show how genes regulating tooth development are ranked high by DELocal even if their expression level changes are subtle. We propose that developmental biology studies can benefit from gene neighbourhood analyses in the detection of differential expression and identification of organ specific genes.

## Introduction

Temporal and spatial regulation of gene expression is important in development and differentiation. During development, genes are expressed in a highly dynamic manner, and perturbations of the expression dynamics underlie many diseases and developmental defects. Regulation of gene expression through developmental time can be examined by quantifying gene expression levels at two or more time points. However, because development is typically a gradual process, detecting differential gene expression can be challenging due to the changes being initially subtle. This challenge underscores the continuing need for different strategies to identify biologically meaningful changes in gene expression.

The relationship between gene expression and genome organization is one aspect that has attracted considerable research interest. In prokaryotes, the genome structure and the regulation of gene expression are operationally linked. This is because the genes involved in the same process are typically located in the same operons, forming multigene clusters that were originally read as single transcripts [1]. In eukaryotes, operons controlling multiple genes are rare although nearly 15% of the 20,000 genes in *C. elegans* genome are located in operons [2,3]. Nevertheless, most eukaryotic genes are scattered throughout the genome without apparent order related to expression or function [4]. Even though there does not appear to be any simple logic in the distribution of eukaryotic genes, it is not entirely random as most genes cannot be moved around in the genome without seriously disturbing their functionality [5]. Some non-random distribution is found both at the small scale, relating to a limited number of genes, and at the large scale, concerning large chromosomal regions [5,6]. Clustering has been detected among co-expressed genes, such as the housekeeping genes, as also within functional groups, such as interacting pairs of proteins or biological processes containing thousands of genes [7,8]. Overall, factors contributing to the genome organization include at least evolutionary history and current evolutionary forces, mechanisms of rearrangement, mechanisms of regulation of gene expression and control of chromatin composition. The main source of non-random gene order in mammalian genomes has been suggested to be tandem duplications [9].

An additional factor in the eukaryote nucleus is the three-dimensional organization of the genome [10,11]. During metazoan development, enhancers regulate the gene activity in different cell and tissue types through dynamic three-dimensional hubs, in which enhancers and promoters assemble in physical proximity to activate gene expression. Genome-wide analyses of the spatial organization of chromatin have uncovered multiple topologically associating domains (TADs) that form largely self-interacting regions of regulatory elements and promoters [12,13]. Although there are cases where gene expression is co-expressed within TADs, [14,15] current evidence points towards a more complex relationship between gene co-regulation and TADs [16,17]. In addition, TADs appear to have multiple hierarchies [16], and questions remain on the degree of TAD-conservation across species [18].

In this study we examine genome organization from the point of individual organ development; the common focus of developmental biology and regeneration studies. We first review

the spatial distribution of protein coding genes in the mammalian genome and quantify neighbouring genes involved in development of the same organs. Next, our specific organ system for in-depth analyses of gene neighbourhoods and gene expression is the mammalian tooth. Especially developing mouse molar teeth provide many well characterized genes to examine links between gene neighbourhood and expression [[19] and references therein]. Our starting point is 1 Mb chromosomal neighbourhood around each organ specific gene as these encompass the majority of both regulatory regions and constrained architecture seen in the conservation of large or very orderly gene clusters. The choice of 1Mb window is obviously somewhat arbitrary, but it should provide a readily applicable framework to diverse taxa and data. This definition of neighbourhood should be fairly compatible with TADs as they happen to be approximately 1Mb in the mouse (Dixon et al. Nature, 485, 2012; they say 880kb)[12]. Nevertheless, we also test our inferences using TADs calculated for each neighbourhood. The 1Mb windows include well known gene clusters such as Hox genes and immunoglobulin genes, whose spatial organization on the chromosome is crucial to their regulation and function. Still, this strict co-localization of co-regulated genes is not the general pattern in mammalian genomes [9], and the patterns of gene clustering related to gene function are delicate and complex.

Finally, for detection of differentially expressed genes in organ development, we developed a new method, called DELocal. This method does not depend solely on the read counts from transcriptome sequencing, but it includes in the analysis the expression data of the neighbouring genes. This approach removes the emphasis on fold-changes that is common to other methods and allows DELocal to identify relatively subtle gene expression changes during development. Using embryonic mouse dental transcriptomes obtained with both microarray and RNAseq, we show how DELocal provides an alternative ranking of differentially expressed genes.

## Results

### Majority of protein coding genes have neighbouring genes within 1 Mb window

As our focus of interest is developmental regulation, we first obtained an overall pattern of distribution of developmental genes by tabulating how closely genes are located in the genome. For example, the human genome (GRCh38.p13) is $4.5^*10^9$ base pairs long and has 20,449 protein coding genes, which means, on average, one gene for every 220,060 bases (Ensembl release 99). Similarly, for the mouse genome, this number is 154,290 (GRCm38.p6). Consequently, on average, 5 to 6 genes should reside in each 1 Mb window in the mouse genome. To express these statistics as a neighbourhood, we can state that each gene has 4 to 5 neighbouring genes within a 1 Mb window. This observation is obviously a broad generalization, but it does indicate that genes tend to have some neighbours within 1 Mb. To examine the neighbourhood patterns in more detail, we examined the 1 Mb neighbourhoods of protein coding genes of the mouse genome. For every gene, number of neighbouring genes within 1 Mb window was counted (Figs 1A and S1). This simple calculation shows that, at the level of all protein coding genes, majority of genes in mouse genome have more than 4 neighbours, the median number of neighbours being 15 (S2 Fig). This tabulation indicates that there is some level of clustering of genes in the eukaryotic genome, a pattern well established in the literature [20].

### Majority of protein coding genes lack neighbours regulating the same organ

To contrast the genome-wide pattern of gene neighbourhoods with that of a single organ system, we first tabulated gene neighbourhoods associated with the development of the mouse

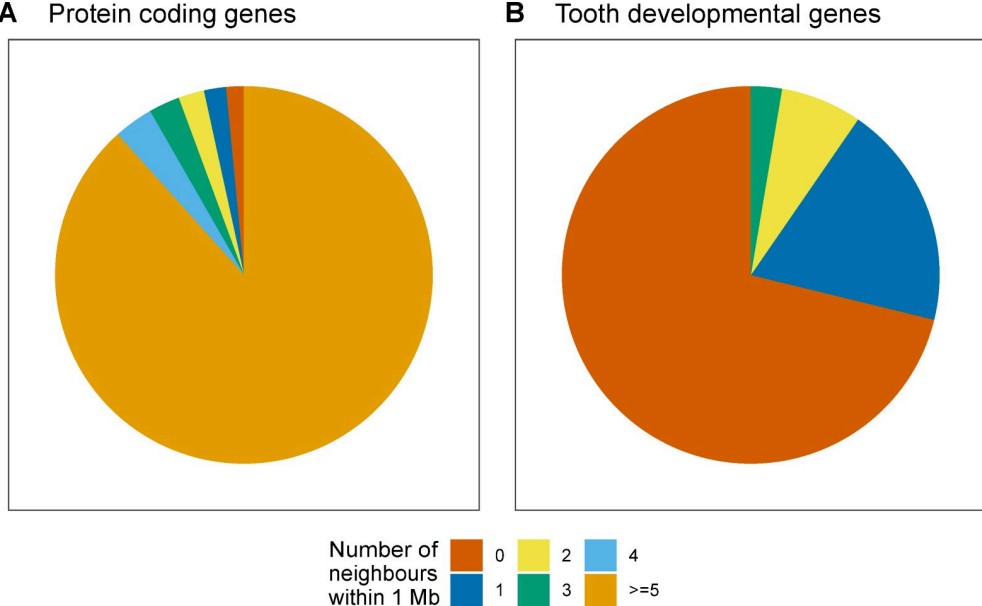

**Fig 1. Although protein-coding genes typically have neighbours, tooth genes do not have other tooth gene neighbours within 1 Mb windows around each gene.** (**A**) The number of neighbouring genes within 1 Mb window around each gene tabulated from the mouse genome for all protein coding genes and (**B**) genes involved in tooth development. Majority of 21,971 protein coding genes have at least five neighbours whereas most of 302 tooth developmental genes do not have tooth genes as neighbours. This pattern suggests that genes with specific functions are sparsely distributed.

tooth. This specific organ focus basically illustrates whether genes participating in the regulation of the same organ are also located close to each other (which is not expected). Here mouse molar development provides a good example because its gene regulation is relatively well understood, and because tooth development itself is a relatively autonomous process [19,21]. We analyzed the 1 Mb neighbourhoods of the tooth developmental genes (S1 Table) and the number of tooth genes that are sharing the same neighbourhood. The results show that tooth genes are mostly located far from each other, suggesting that genes regulating this specific developmental process are not co-localized (Fig 1B), a pattern that is in agreement with the organization of eukaryotic genomes [10,11]. Taken together, although majority of protein coding genes are to some extent clustered in the mouse genome, tooth genes tend to be located far from each other (Fig 1).

Next, we examined how representative the developing mouse tooth (Fig 1B) is of other developmental systems. We used the gene ontology (GO) terms to provide an approximation of the adjacency of genes involved in the same developmental process. Here we focused in the genes belonging to mouse GO slims (a concise list of terms) of "biological process" [22,23]. For every gene belonging to these terms, its 1 Mb neighbourhood was investigated to tabulate genes belonging to the same term. The tabulations show that most genes belonging to a certain GO term are sparsely distributed in the chromosomes (Fig 2) which is in accordance with previous studies [5,24]. There are only few genes from broader, or high level, GO terms that are densely located (more than 3 genes from the same GO term within 1 Mb neighbourhood). However, these few GO terms represent very broad descriptions of biological functions. With more precise GO terms (Fig 2, bottom row), genes tend to have no neighbours belonging to the same GO term. For generality, we examined these patterns in human genome and they remained largely the same (S3 Fig).

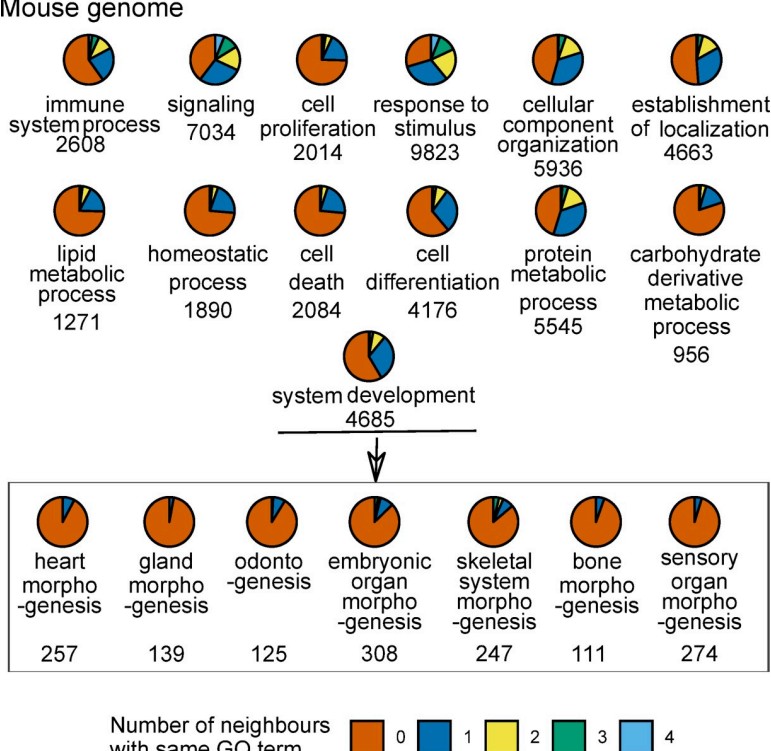

**Fig 2. The more specific the GO term is the fewer neighbours its genes have from the same GO term within 1 Mb.**
Each pie represents the genes of one GO term under the root GO term 'biological process'. The top three rows
represent GO slim terms. The GO terms in the box are children of the GO term 'system development'. The color-
coding indicates the number of neighbours that a gene has from the same GO term. The number of genes in the GO
term is indicated under the GO term. Analysis was done for the mouse genome. See S2 Table for GO IDs.

The scarcity of neighbouring genes in the precise GO term categories is not surprising con-
sidering the limited number of genes in each category. To test if the patterns are a simple func-
tion of group size, we performed simulations where genes were assigned randomly to artificial
GO terms containing different numbers of genes. For every group size, 10,000 simulations
were performed and the density of genes was calculated for each artificial GO term in the 1
Mb neighbourhood. A plot showing the percentage of genes lacking neighbours shows the
expected decrease as the number of genes increases in the artificial GO terms (Fig 3). The
empirical patterns largely follow the randomizations, although some real GO terms with small
number of genes (300 and less) show slightly lower percentages, indicating that there is a slight
tendency for higher spatial clustering of genes belonging to the same GO terms than in the
simulations. This may be partly due to GO terms having paralogous genes that are sometimes
located near each other in the genome. Nevertheless, up to GO term categories containing
1000 genes, 80% of genes have no neighbours belonging to the same GO terms. These patterns
indicate that the general expectation for the genes sharing the same genomic neighbourhood
(Fig 1A) is that they participate in the regulation of different systems and organs. To the extent
that this is the general rule of genomic organization, it provides opportunities for new compu-
tational approaches. Below we develop a method to detect differential expression using addi-
tional information from the neighbouring genes and test it using our data on the developing
mouse tooth.

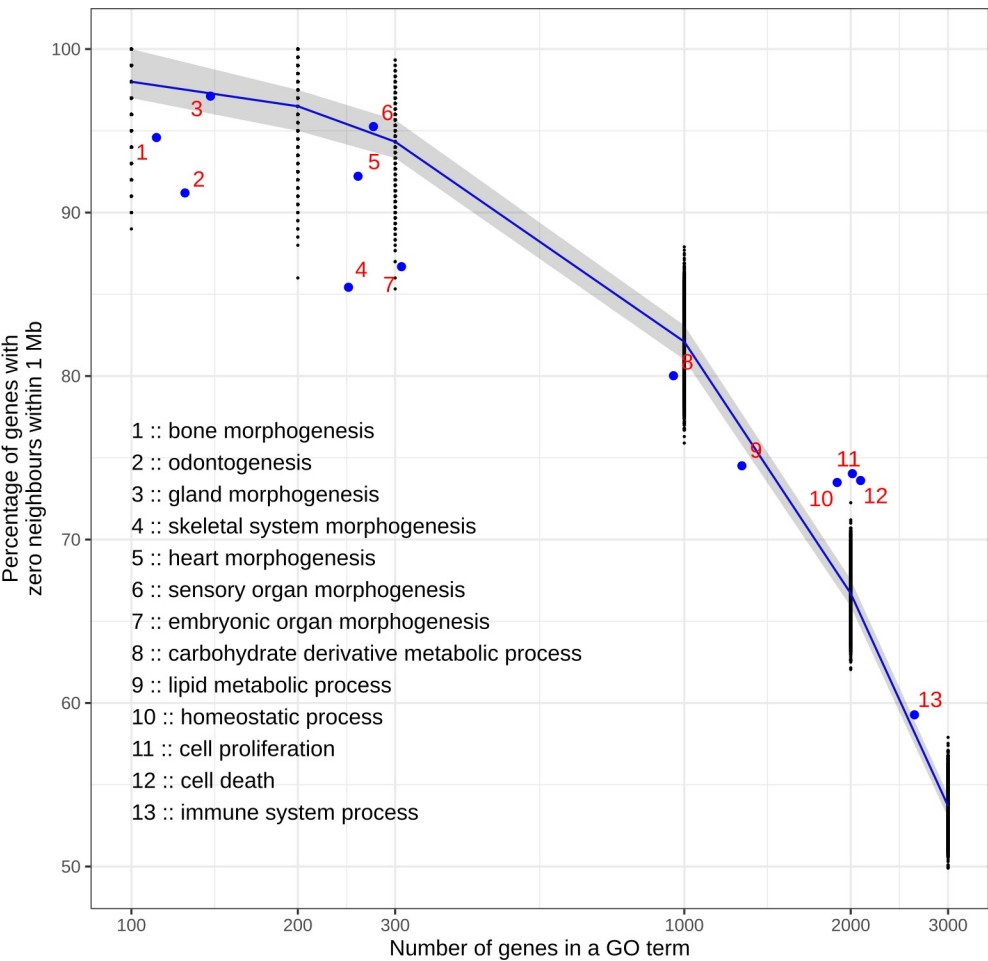

1 :: bone morphogenesis
2 :: odontogenesis
3 :: gland morphogenesis
4 :: skeletal system morphogenesis
5 :: heart morphogenesis
6 :: sensory organ morphogenesis
7 :: embryonic organ morphogenesis
8 :: carbohydrate derivative metabolic process
9 :: lipid metabolic process
10 :: homeostatic process
11 :: cell proliferation
12 :: cell death
13 :: immune system process

**Fig 3. Real GO terms with lower gene numbers are slightly more clustered in the genome than artificial GO terms with the same gene numbers.** Artificial GO terms with different numbers of genes were made with randomly selected genes and their distribution across the genome was measured. For each group size, 10,000 simulations were executed and for each simulation the percentage of genes with zero within-1Mb-neighbours with the same artificial GO term were counted (black dots). Real GO terms are marked with blue dots. See S2 Table for GO IDs. The blue line is median, and the shadow shows the observations between 1st and 3rd quartile.

## Neighbouring genes do not show organ-specific changes in expression

Because genes linked to tooth development do not appear to be co-localized in the genome (Fig 1B), we checked whether these genes differ in their expression dynamics compared to the other genes in their neighbourhood. We examined differential expression of genes at the onset of tooth crown formation, between mouse embryonic day 13.5 (E13) and 14.5 (E14) when many of the tooth genes are known to be upregulated [19,25]. For example, in visual inspection *Ctnna1*, *Shh*, *Foxi3* and *Sostdc1*, all genes required for normal tooth development [26], show upregulation between E13 and E14 (Fig 4). In contrast, the other genes in their neighbourhoods show no marked changes between E13 and E14, irrespective of their expression levels (Fig 4). Therefore, the distribution of genes involved in the regulation of the tooth (Figs 1 and 4) is manifested also at the level of differential expression. Building from this observation, we developed a new algorithm, DELocal, to identify differentially expressed genes based on their neighbours' expression dynamics (Materials and Methods). To evaluate its potential

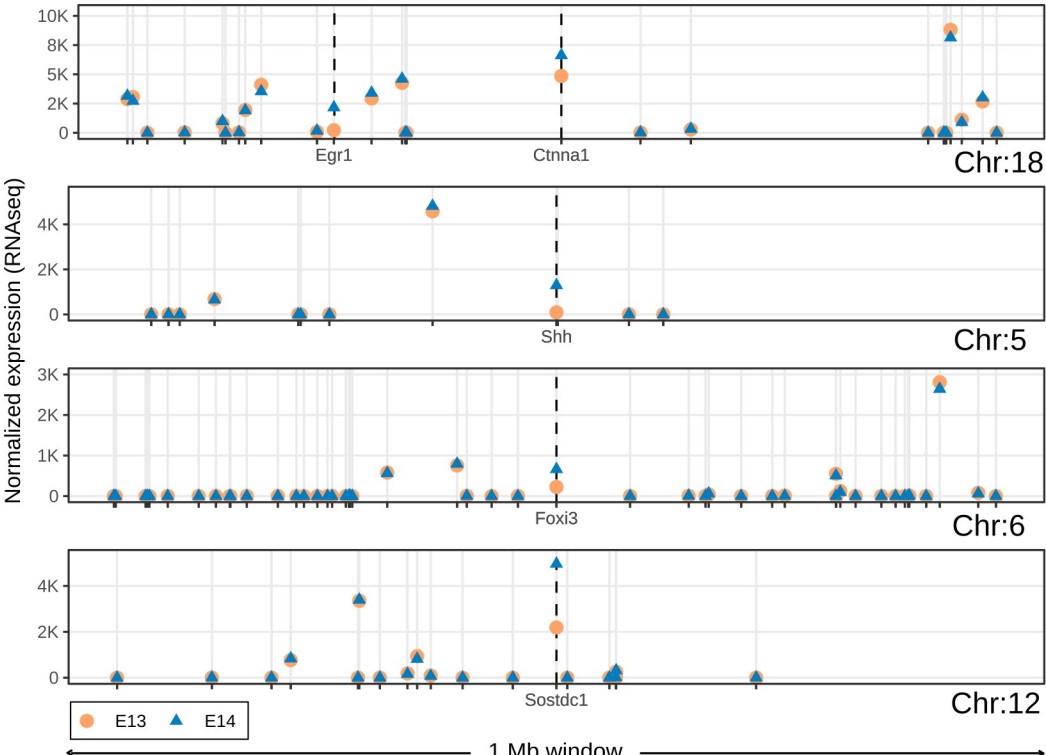

**Fig 4. Only tooth developmental genes are differentially expressed within 1 Mb windows in the developing mouse molar.** Median expression levels of the tooth developmental genes *Ctnna1*, *Shh*, *Foxi3* and *Sostdc1* and their neighbouring genes at developmental stages E13 and E14. Regardless of their expression level, the surrounding genes show little change between the two stages whereas the tooth genes are upregulated. *Egr1* in the 1 Mb window of *Ctnna1* is also a tooth developmental gene.

performance, we used gene expression data from embryonic mouse dental tissues generated by both microarray and RNAseq.

## The number of neighbours is not critical for detecting differential expression

Our hypothesis of neighbouring genes being informative in the detection of differential expression is dependent on the definition of 'neighbourhood'. Therefore, it is important to determine the right number of neighbours to include in the analysis by the DELocal algorithm. To define the optimal number of neighbours we tested different numbers (1–14) of closest genes within a fixed window (1 Mb) surrounding the gene of interest. We evaluated the performance of DELocal with different numbers of closest neighbours in identifying the genes involved in tooth development. Again we contrasted the expression levels between E13 to E14 molar teeth (for 302 tooth genes), or so-called bud stage to cap stage transition in tooth development, when many tooth genes are known to be active [19]. The Matthews correlation coefficient (MCC) scores were measured for different numbers of neighbours to examine the strength of DELocal to identify tooth genes (true positive; TP) as well as non-tooth genes (true negatives; TN). The MCC score was chosen to optimize the model due to very few TPs, or imbalanced dataset. The results show that DELocal produces similar and stable MCC scores on both microarray and RNAseq datasets, even though RNAseq data produces slightly higher MCC scores than microarray (Fig 5). We note that only one nearest gene is enough to obtain

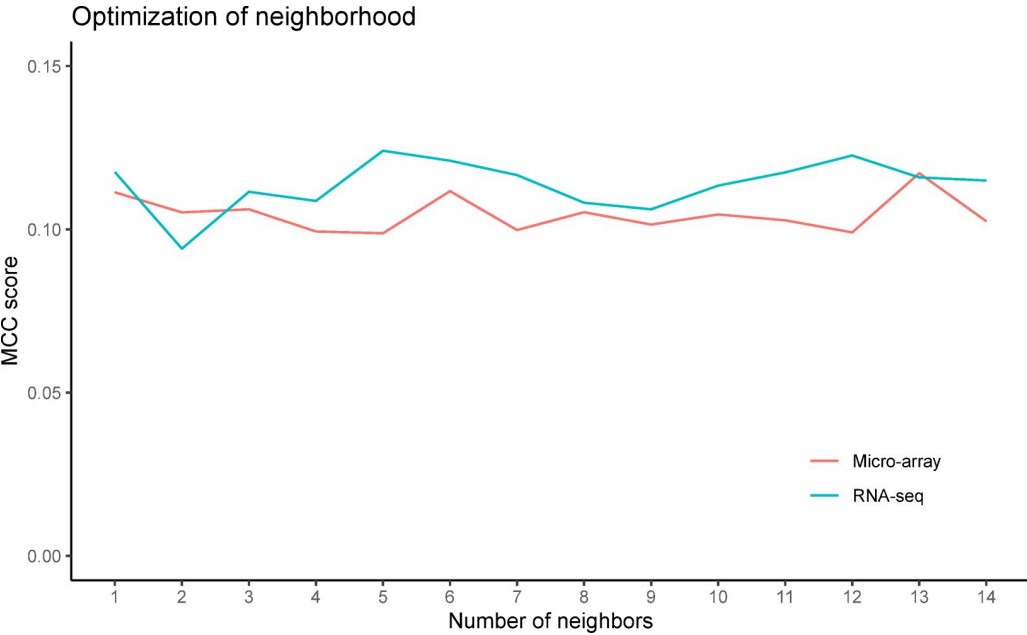

**Fig 5. DELocal performance is not strongly dependent on the number of gene neighbours used in the analysis.** Every gene is evaluated in relation to its neighbouring genes. In the absence of any "gold standard" for the number of neighbours, different numbers of nearest genes (within 1 Mb window) were used to identify the differentially expressed genes. The overall performances were measured using MCC. The performance of DELocal using RNAseq data was slightly better than with microarray data.

close to the highest MCC score. However, for RNAseq the best MCC score corresponds to 5 nearest neighbours. Because there are fewer genes available in microarray analyses compared to RNAseq, in the rest of the analyses we used DELocal with 5 neighbours both for the RNAseq and microarray data. It should be also noted that the distances of these 5 neighbours are different for each gene and therefore the final neighbourhood windows are not constant in size. Additionally to the 1 Mb window, median TAD boundaries were also used to define the neighbourhoods and to evaluate DELocal. Although, TAD boundaries are different for each gene there was no marked difference in the result compared to 1 Mb neighbourhood (S4 Fig).

## DELocal shows increased precision compared to other methods

Microarray is one of the earliest successful high throughput technologies to measure a large number of gene expressions, and consequently there are a good number of statistical methods to identify differentially expressed genes from datasets generated by this platform. Hence, the performance of DELocal can be evaluated by comparisons to these methods using a microarray dataset. However, microarray is limited to only the genes that have been targeted by microarray probes. Therefore, the expression of all the genes cannot be accessed, resulting in fewer neighbouring genes being sampled. To obtain a more comprehensive readout of differentially expressed genes, RNAseq was also used to evaluate DELocal performance. The performance of all these methods was measured by their ability to identify differentially expressed tooth genes.

For analysis of performance, we used the receiver operating curve (ROC) [27] depicting the true positive rate against the false positive rate of differentially expressed genes. The analyses show that DELocal outperforms other methods in identifying tooth genes using both microarray and RNAseq (Fig 6). The performance is most similar to limma/DEMI (microarray) and limma/DESeq (RNAseq). We used also other metrics like specificity, recall (sensitivity),

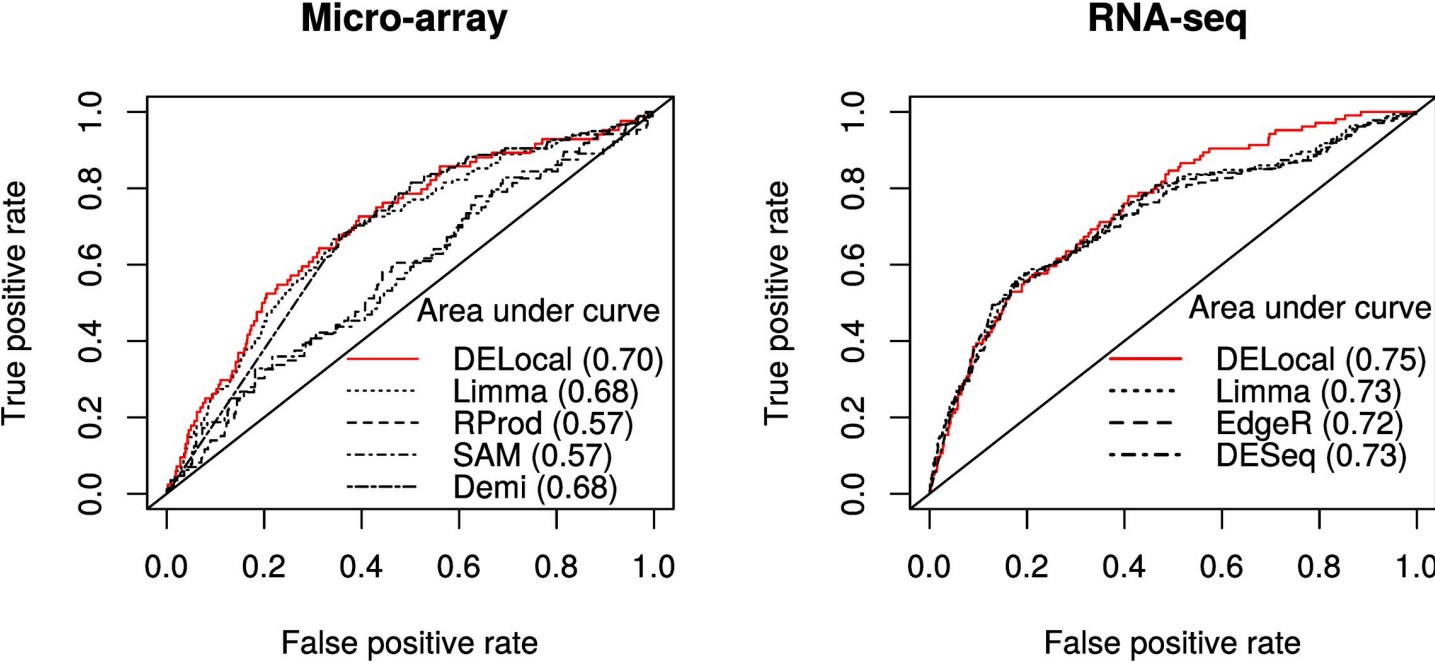

**Fig 6. Compared to other methods, DELocal is at least as powerful in detecting differentially expressed genes.** Receiver operating characteristic (ROC) curves and areas under the curves (within the parenthesis) show that DELocal outperforms other methods on both microarray and RNAseq data.

precision and MCC to evaluate and compare the different methods. DELocal shows high specificity and accuracy compared to other methods for microarray data (S5 Fig).

In RNAseq data, DELocal outperforms other methods except in recall (sensitivity) (Fig 7). The MCC scores remain equivalent to each other. The tooth gene dataset is imbalanced due to the large number of non-tooth genes (true negatives), which hinders the evaluation of accuracy, but does not affect F1 or MCC. Considering that the objective of many experiments is to find true positives, the F1 score, which is a compound-term of precision and recall, is an important metric. The F1 score ranges from zero (bad) to one (good), and the F1 scores of all of the methods remain suboptimal. Nevertheless, DELocal using RNAseq outperforms other methods also in F1 score. Below we discuss the results from RNAseq only.

### DELocal provides an alternative ranking of genes by removing bias towards large changes in expression

Because DELocal quantifies expression changes in the context of neigbouring genes, genes in the chromosomal regions having generally large changes in expression should not be as highly ranked as they would be in other methods. We examined the rankings and expression changes of tooth genes among predicted differentially expressed genes by all the methods. DELocal predicted tooth genes are enriched in top rank positions compared to the other three methods (S6 Fig). In experimental research, gene rankings are typically the first values to be examined in validating the results biologically, and in practice high ranking is very influential when selecting candidate genes for new downstream analyses. The results show that whereas DEseq, edgeR and limma emphasize in their rankings genes that show large changes in their expression, this bias is not present in the DELocal ranking (Fig 8A). These results also indicate that the identity of highly ranked tooth genes is different between DELocal and the other methods (Fig 8B). Thus, DELocal differs from the other methods more substantially than is apparent based on the different performance measures alone (Figs 6 and 7).

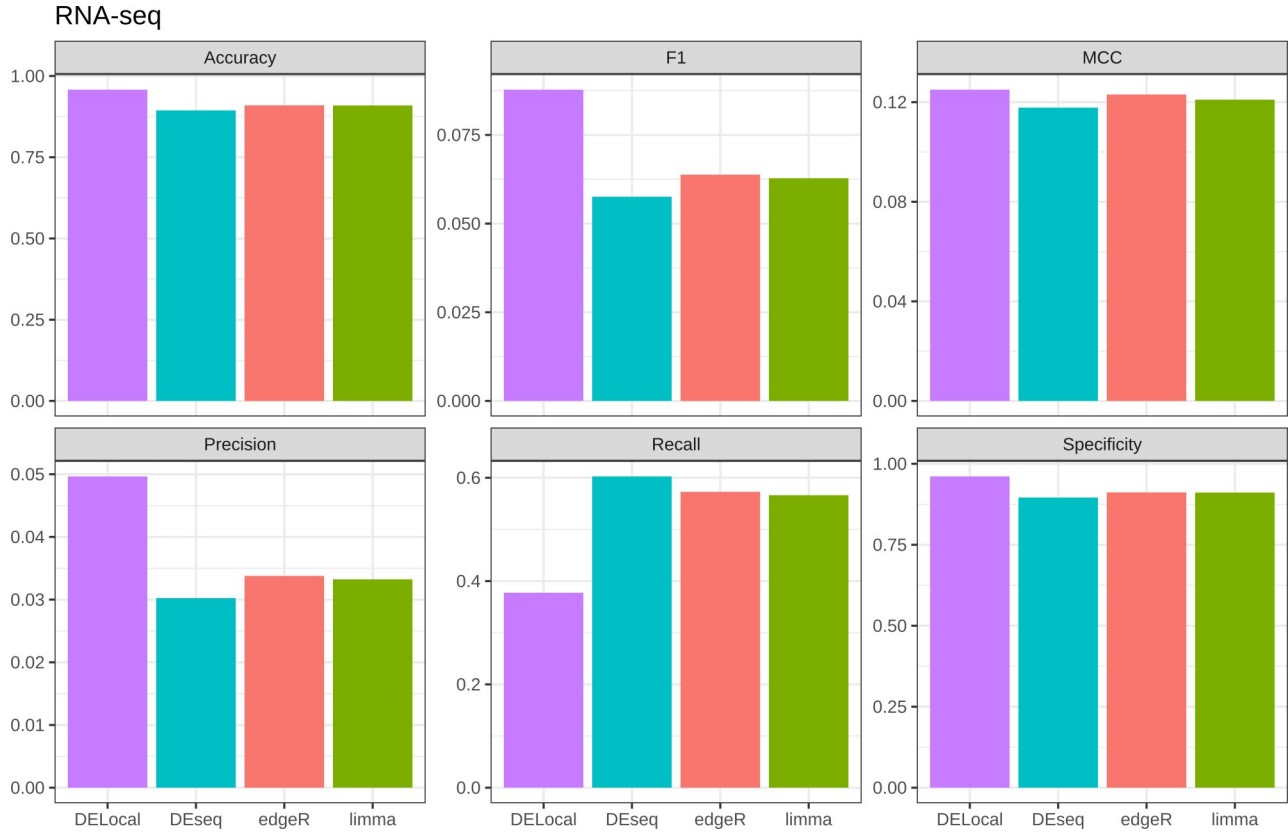

**Fig 7. Comparison of DELocal with earlier methods to identify differential expression.** Evaluation matrices show that, except for recall (sensitivity), DELocal equals or outperforms earlier methods. However due to large number of true negatives, the significance scores of precision, F1 and MCC remained negligible. The evaluation matrices are explained in Materials and Methods section. The analysis was done using RNAseq data. For microarray data see S5 Fig.

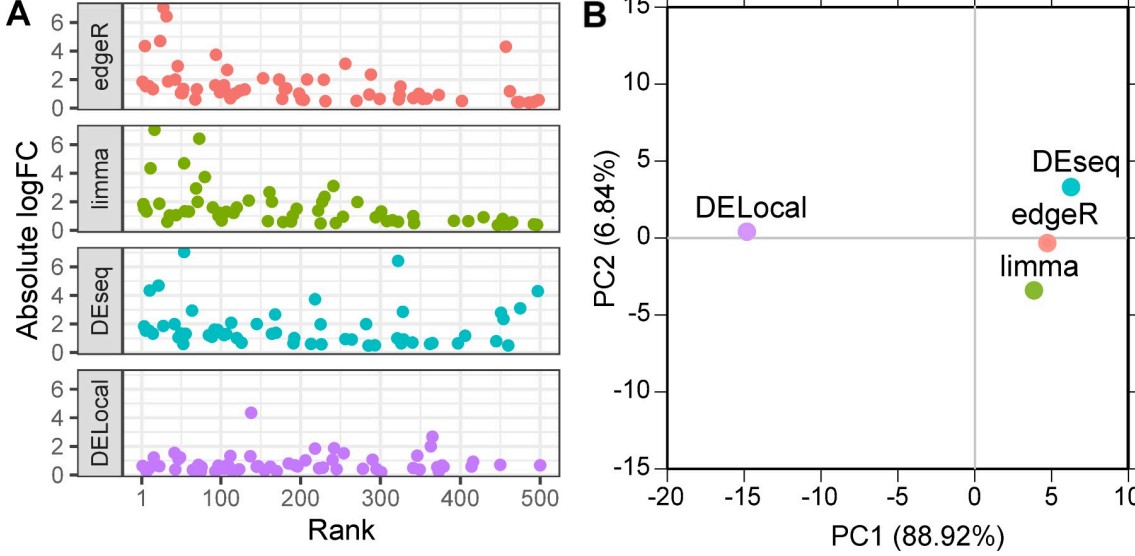

**Fig 8. DELocal provides an alternative ranking of differentially expressed genes.** (A) Unlike the other methods, DELocal rankings of differential expression do not bias towards genes with large log-fold changes (1 denotes the highest rank). (B) Principal component analyses of the rank orders of tooth developmental genes shows the ordering of the DELocal to be distinct from the ones provided by the other three methods (using correlation matrix of top 500 tooth genes).

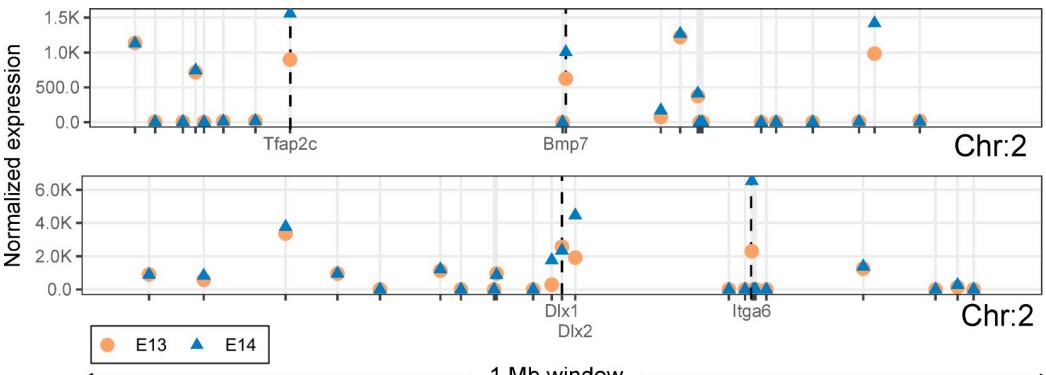

**Fig 9. Differentially expressed genes in the same neighbourhood interfere with the detection of differentially expressed genes by DELocal.** DELocal failed to identify *Bmp7* due to differential expression patterns of its neighbouring genes (top). *Dlx1* and *Dlx2* are paralogous tooth genes which are in the same neighbourhood (bottom). Only tooth developmental genes are labelled here.

## Differentially expressed genes missed by DELocal tend to have local paralogues

The DELocal algorithm appears to efficiently identify the differentially expressed genes (having high precision) as well as to filter out non-tooth genes (having high specificity). Still, DELocal missed 60 differentially expressed tooth genes which are identified by all the other methods in RNAseq dataset. DELocal is built on the hypothesis that every true differentially expressed gene should have neighbours and none of them should be differentially expressed. Consequently, DELocal may fail to identify those differentially expressed genes whose neighbours are also differentially expressed. For instance, 2 out of 5 nearest neighbours of *Bmp7* were differentially expressed genes which could be the reason of failure of DELocal to detect *Bmp7* (Fig 9). Additionally, the presence of paralogous genes in the neighbourhood may complicate the detection of differential expression. Most notably, *Dlx1* and *Dlx2* in chromosome 11 (Fig 9), *Dlx5* and *Dlx6* in chromosomes 6, and *Dlx1* and *Dlx2* in chromosome 2, are co-regulated as pairs and can compensate for the deletion of one another [28–31]. Additionally, the other neighbouring paralogs in our set of tooth genes are *Cyp26c1* with *Cyp26a1*, and *Cdh1* with *Cdh3* [32]. Therefore, at least some of the genes missed by DELocal should be possible to detect by incorporating information about gene paralogues into the algorithm.

## Discussion

Only seven protein-coding genes in Ensembl mouse annotation lack neighbours altogether within their 1 Mb windows (Fig 1A). Hence, the overwhelming majority of mouse genes have neighbours, and this pattern of organization is likely to apply to all mammals, if not beyond. In the genomic scale of thousands of genes, neighbouring genes can be co-expressed in eukaryotes [33]. Our analyses, focused on individual organ development, suggest that the neighbouring genes are very rarely involved in the regulation of the same organ or tissue (Fig 2), and that this is also reflected in the lack of co-expression of neighbours (Fig 4). The lack of co-expressed neighbours is partly explained by the limited number of genes that are associated with individual organ regulation (Figs 2 and 3). In contrast, more general processes, such as cell proliferation or cell death, are linked to many thousands of genes (Fig 2), increasing the likelihood of neighbourhood co-expression. Indeed, an analysis using high-level GO terms containing thousands of genes found that functionally related genes are often located close to each other [8]. It

is plausible that evolution of organs such as teeth may have co-opted genes from multiple high-level clusters, perhaps further explaining the rarity of co-expressed neighbours in organ regulation.

A distinct advantage of the rarity of co-expressed neighbours in organ development is that this information can be used to detect even subtle changes in gene expression. To this end, here we developed the DELocal algorithm to identify differentially expressed genes by incorporating expression information from chromosomal neighbourhood. DELocal provided highly precise detection of differentially expressed genes in mouse tooth development (Figs 6 and 7), suggesting that this method can provide additional benefits in the analysis of developmental systems. Specifically, DELocal is distinct from the previous methods of determining differentially expressed genes by not relying on large fold-changes in gene expression (Fig 8A). The ability of DELocal to identify biologically significant but more subtle changes is based on its special feature: it can track a change in gene expression that is unique among the surrounding genes in the chromosomal location. Thus, DELocal can pinpoint genes which are being actively and specifically regulated between two developmental stages, regardless of whether the change of expression is large or small. Taken together, DELocal can complement other methods by providing alternative rank listings of potential genes of interests (Fig 8B).

More technically, the developed DELocal algorithm is based on linear models that have been successfully implemented and used in limma, DESeq2 and many other methods to identify differentially expressed genes [34,35]. With linear models, gene expression can be modelled in two or more biological conditions and thereafter differentially expressed genes of different contrasts of interest can be found. Linear models are advantageous compared to other methods in that they can model complex experimental conditions with multiple factors. Here we used DELocal to determine the genes that are differentially expressed between bud and cap stage in the developing mouse tooth. The extensive list of genes active in tooth development [19,21] allowed us to optimize and evaluate the performance of DELocal. Considering that the *in vivo* bud and cap stage differences in gene expression are relatively subtle, the high specificity and accuracy of DELocal is promising. Obviously, the optimization requires a list of genes of interest. However, DELocal can also be run without any prior knowledge of genes that are active in a particular developmental process. Fig 3 shows that in relation to the number of neighbours, the performance of DELocal is very stable and even only one nearest neighbour could be sufficient to build the models. The implication of this is that DELocal could be used with only a single neighbour contrast in the absence of any reference/training gene set. In this context it will be interesting to test whether the optimum number of neighbours is the same for different organs. Another future direction is to examine different TAD classifications using DELocal.

It remains to be explored whether the rarity of neighbouring genes regulating the same process is related, for example, to the requirements of spacing required for folding of the DNA, or whether it is the result of historical processes of genome rearrangements or co-option of genes in the evolution of organs. Regardless, our results on the developing mouse tooth suggest that the lack of organ specific neighbours is also manifested at the level of differential expression. We further show that it is possible to develop computational approaches to detect even subtle changes in gene expression using the additional information from the neighbouring genes.

## Materials and methods

### Ethics statement

All mouse studies were approved and carried out in accordance with the guidelines of the Finnish national animal experimentation board under licenses KEK16-021, ESAVI/2984/04.10.07/2014 and ESAV/2363/04.10.07/2017.

## Gene sets and gene neighbourhood analyses

The start position, chromosome name and gene biotype (protein coding, lncRNA, ncRNA, pseudogene) of every gene of mouse and human genomes were downloaded from Ensembl database using R package biomaRT [36] in February, 2020. Further analysis was limited to protein coding genes. 302 genes linked to tooth development were marked as tooth developmental genes (S1 Table and refs [19,37]). Throughout the study, the gene start coordinates are used as the position of the gene in the chromosome. The same coordinates are used to measure the distances between genes. The GO slim terms were acquired from AmiGO [38] and they are listed in S2 Table. For each GO term, the corresponding genes were downloaded again from Ensembl using biomaRT. The artificial GO terms of different sizes for simulation were created by randomly sampling protein coding genes from genome using custom R scripts. For every group size, 10,000 simulations were performed and for each simulation density of genes from the same artificial GO term in neighbourhood was calculated.

For most of the analyses, we used the 1 Mb windows as a neighbourhood measure for each focal gene as these are relatively straightforward to determine for different genomes and genes. Additionally, we analyzed neighbourhoods using TADs. Although TADs are considered to be conserved across different mammalian cell types, multiple TADs surrounding a gene with different boundaries can be delineated [39]. For a robust consensus of TAD boundaries, we downloaded four different mouse cell-lines and three different algorithms (with 10 kb and 50 kb resolutions, June 2021 [40]). A consensus TAD boundary for each gene was defined by selecting a median start and end co-ordinates from all TADs surrounding that gene. The median size of these consensus TAD boundaries was 990kb (within the range of 185kb to 6Mb, standard deviation = 547813.8) which is roughly similar to previous report of 880kb [41]. TAD boundaries for more than 1500 genes (mostly from Y chromosome) were not determined.

## RNAseq library preparation

Developing mouse molar teeth from embryonic days 13.5 (E13) and 14.5 (E14) were dissected from wild type C57BL/Ola embryos. For RNAseq, five biological replicates were used. The samples were stored in RNAlater (Qiagen GmbH, Hilden, Germany) in -75˚C. RNA was extracted first twice with guanidinium thiocyanate-phenol-chloroform extraction and then further purified using RNeasy Plus micro kit (Qiagen GmbH, Hilden, Germany) according to manufacturer's instructions. RNA quality of representative samples was assessed with 2100 Bioanalyzer (Agilent, Santa Clara, CA) and the RIN values were 9 or higher. The RNA concentrations were determined by Qubit RNA HS Assay kit (Thermo Fisher Scientific, Waltham, MA). The cDNA libraries were prepared with Ovation RNAseq System V2 (Nugene, Irvine, CA), and sequenced with NextSeq500 (Illumina, San Diego, CA).

## Microarray library preparation

Mouse E13 and E14 teeth were dissected from wild type NMRI embryos. Five biological replicates were used. The amount of RNA available in each sample was measured with 2100 Bioanalyzer (Agilent, Santa Clara, CA). Only the samples showing a RIN value above 9 were used for the microarray analysis.

## Gene Expression analysis

Gene expression was measured both in microarray (platform: GPL6096, Affymetrix Mouse Exon Array 1.0) and RNAseq (platforms GPL19057, Illumina NextSeq 500). The microarray

gene signals were normalized with aroma.affymetrix [42] package using Brainarray custom CDF (Version 19, released on Nov 13, 2014) [43]. The RNAseq reads (84 bp) were evaluated and bad reads were filtered out using FastQC [44], AfterQC [45] and Trimmomatic [46]. This resulted in on average 63 million reads per sample. Then good reads were aligned with STAR [47] to Mus_musculus.GRCm38.dna.primary_assembly.fa genome and counts for each gene was performed by HTSeq [48] tool using Mus_musculus.GRCm38.90.gtf annotation. On average 89% reads were uniquely mapped to *Mus musculus* genome. Additionally, RNAseq count values were normalized using DESeq2 [34]. All the transcriptomic data are available in NCBI Gene Expression Omnibus under the accession number GSE142201.

## DELocal

In DELocal, it is hypothesized that differentially expressed genes have different expression dynamics compared to their neighbouring genes. Specifically, we want to find genes whose expression changes, while that of the surrounding genes do not. To calculate this, we used a similar logic to ESLiM [49], an algorithm that detects changes in exon usage. Whereas ESLiM is suitable for detection of alternative splicing (exon usage), DELocal is applicable for identifying differentially expressed genes in the chromosomal neighbourhood. Note that in principle one could choose the neighbouring gene or genes randomly across the genome, but this would result in different results in each randomization.

In this algorithm, gene's expression is modelled as a linear relationship with median expression of neighbourhood genes, such as,

$$\widehat{g}_{ij} = s_i \times N\tilde{g}w_{ij} + b_i, \tag{i}$$

Where $\widehat{g}_{ij}$ is expected expression of $i$-th gene in $j$-th sample, $N\widehat{gw}_{ij}$ is median expression of $N$ nearest neighbouring genes within 1 Mb window of the $i$-th gene from $j$-th sample and $b_i$ is base line expression level of gene $g_i$. The slope $s_i$ of every gene $g_i$ depends on its neighbouring genes. Therefore, the difference between expected and observed values or residual,

$$r_{ij} = g_{ij} - \widehat{g}_{ij} \tag{ii}$$

where $g_{ij}$ is observed value. For differentially expressed genes, these residual values will be significantly different in different biological conditions.

Furthermore, with the aid of the residuals $r_{ij}$ observed $g_{ij}$ could be formulated as follow,

$$g_{ij} = s_i \times N\tilde{g}w_{ij} + b_i + r_{ij} \tag{iii}$$

Noticeably, this relationship Eq (iii) is independent of experimental condition and only dependent on neighbouring gene. Therefore, differentially expressed genes are detected through significantly deviated residual values between the desired contrasts using Empirical Bayes statistics, available from limma package [35]. We tested the performance of DELocal using from 1 to 14 neighbouring genes ($N$ in Eq (iii)). The log-normalized and normalized count values were used in DELocal respectively for microarray and RNAseq data. There are 334 protein coding genes in mouse genome which do not have any other protein coding gene in their 1Mb neighbourhood. Therefore, we also used available non-protein coding genes from the neighbourhood in DELocal analysis. However, after the inclusion of non-protein coding genes there are still 17 protein coding genes without any neighbours within 1 Mb.

## Performance measures

DELocal was compared with different publicly available tools applicable both for microarray or RNAseq: RankProd [50], SAM [51], DEMI [52], limma [35], edgeR [53] and DESeq2 [34]. All these programs were executed with default parameters. Genes reported with p-value < = 0.05 by these tools were marked as differentially expressed and used to evaluate the performance of each tool using the following metrics and receiver operating characteristic (ROC) curves.

- Sensitivity (Recall), true positive rate TPR = TP/ (TP + FN)

- Specificity, true negative rate SPC = TN/ (TN + FP)

- Precision, positive predictive value PPV = TP/ (TP + FP)

- Accuracy, ACC = (TP + TN) / (TP + FP + FN + TN)

- F1 score, F1 = 2TP/ (2TP + FP + FN)

- Mathews correlation coefficient, $MCC = \frac{TP \times TN - FP \times FN}{\sqrt{[(TP+FP)(TP+FN)(TN+FP)(TN+FN)]}}$

where TP, true positive; FP, false positive; TN, true negative; FN, false negative.

Tooth developmental genes were used to find the true and false positive rate for the analyses (S1 Table and refs [19,37]). Rest of the genes (in thousands) were considered as true negatives. Due to the large number of these, very high accuracy can be achieved by predicting all genes as non-differentially expressed. Therefore, because our objective was to find true tooth genes, we focused on Recall, Precision, F1, MCC and area under the ROC curves rather than only Accuracy for evaluating the methods. The areas under the ROC curves were calculated with ROCR [27] package. Additionally, we compared differences in rank and log-fold changes in the genes ranked high by the different methods (Fig 8).

## Supporting information

**S1 Fig.** Distribution of genes across mouse chromosomes (x-axis) and number of "neighbouring genes" (y-axis) within 1 Mb window around each gene.
(TIF)

**S2 Fig. Genes with 5–9 neighbouring genes are most frequent in the mouse genome.** Each point represents the number of genes with the same number of neighbouring protein coding genes in 1 Mb window.
(TIF)

**S3 Fig. The more specific the GO term, the fewer neighbours its genes have.** Each pie represents the genes of a GO term under the root GO term 'biological process'. The GO terms are arranged from top to bottom following their proximity to the root term 'biological process'. The color coding indicates the number of neighbours that a gene has from the same GO term. Analysis was done for the human genome. See S2 Table for GO IDs.
(TIF)

**S4 Fig. Performance comparisons of DELocal calculated using neighbourhoods with 1 Mb windows and with TAD boundaries (with RNAseq data).**
(TIF)

**S5 Fig. Comparison of DELocal with earlier methods to identify differential expression.** Evaluation matrices for microarray data. DELocal has lower performance in microarray data

compared to RNAseq data (Fig 7), likely to be due to limited number of genes and neighbourhood data in microarrays. The evaluation matrices are explained in Materials and Methods section.
(TIF)

**S6 Fig. Distribution of ranks of tooth developmental genes.** DELocal predicted tooth developmental genes are significantly (Wilcoxon rank test; p-value < = 1.0e-06) enriched in top ranked positions compared to the other three methods. Although other methods identified more of the genes (numbers listed next to the box plots), this improved recall is sacrificing specificity. DELocal balances both which is also reflected in F1, MCC and ROC curves (Figs 6 and 7).
(TIF)

**S1 Table. List of tooth developmental genes.**
(XLSX)

**S2 Table. List of GO terms analysed in Figs 7, 8 and S4.**
(DOCX)

## Acknowledgments

We thank P. Rastas and P. Auvinen for discussions. We thank P. Auvinen, L. Paulin and P. Laamanen at DNA Sequencing and Genomics Laboratory for bulk RNA sequencing.

## Author Contributions

**Conceptualization:** Rishi Das Roy, Outi Hallikas, Jukka Jernvall.

**Data curation:** Rishi Das Roy, Outi Hallikas.

**Formal analysis:** Rishi Das Roy.

**Funding acquisition:** Jukka Jernvall.

**Investigation:** Rishi Das Roy, Outi Hallikas, Mona M. Christensen, Elodie Renvoisé.

**Methodology:** Rishi Das Roy, Outi Hallikas.

**Project administration:** Rishi Das Roy, Outi Hallikas, Jukka Jernvall.

**Software:** Rishi Das Roy.

**Validation:** Rishi Das Roy, Outi Hallikas.

**Visualization:** Rishi Das Roy, Outi Hallikas, Jukka Jernvall.

**Writing – original draft:** Rishi Das Roy, Outi Hallikas, Jukka Jernvall.

**Writing – review & editing:** Rishi Das Roy, Outi Hallikas, Mona M. Christensen, Elodie Renvoisé, Jukka Jernvall.

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
