## [Decision Letter · Decision Letter 0]

10 Jun 2021

Dear Das Roy,

Thank you very much for submitting your manuscript "Chromosomal neighbourhoods allow identification of organ specific changes in gene expression" for consideration at PLOS Computational Biology.

As with all papers reviewed by the journal, your manuscript was reviewed by members of the editorial board and by several independent reviewers. In light of the reviews (below this email), we would like to invite the resubmission of a significantly-revised version that takes into account the reviewers' comments.

We cannot make any decision about publication until we have seen the revised manuscript and your response to the reviewers' comments. Your revised manuscript is also likely to be sent to reviewers for further evaluation.

Sincerely,

Alexandre V. Morozov, Ph.D.

Associate Editor

PLOS Computational Biology

Sushmita Roy

Deputy Editor

PLOS Computational Biology

Reviewer's Responses to Questions

**Comments to the Authors:**

Reviewer #1: 1) This manuscript first assesses the distribution of genes related to specific biological functions and organs. The functions were defined by gene ontology terms or experimental data from tooth development studies. It was found that even though most genes have several neighbors within 1 Mb, the neighboring genes rarely participate in the same function as might be expected if the bacterial operon control mode were followed in these systems.

It is well-known, however, that gene regulation in animals occurs by entirely different mechanisms than in bacteria, using expression hubs that bring together for coordinate expression linearly distant genes. So, although the results provided in this part of the paper are significant, they are not unexpected. One 18-year-old paper (ref. 19) that hints at this explanation is cited on p. 6, but it is not enough to dispel the impression that the result is surprising.

The authors have an excellent statement relating to this in their Discussion: “It remains to be explored whether the rarity of neighbouring genes regulating the same process…folding of the DNA, or whether it is the result of historical processes of genome rearrangements or co-option of genes in the evolution of organs.” But these ideas should also be raised in the Introduction in my opinion, and more recent treatments such as https://pubmed.ncbi.nlm.nih.gov/30262496/ or https://pubmed.ncbi.nlm.nih.gov/31226276/ should be cited.

2) The paper then advances a new method, DELocal, to determine the relative activity of genes in chromosomal regions within a 1 Mb window in animal cells. Expression of genes during tooth development is used as an index case of coordinate expression in organogenesis.

The authors compare the performance of their DELocal method on microarray and RNASeq data (the latter being more abundant) to other methods for evaluating differential gene expression: SAM, DEMI, Limma, edgeR and DESeq2. The measures used were Sensitivity, Specificity, Precision, Accuracy, F1 score, and Mathews correlation coefficient. While these terms of art in the gene expression field are defined by algebraic expressions in the Methods section, they seem important enough for them to also be better described in the main text for the benefit of nonspecialist developmental biologists and others who would be interested in the qualitative implications of the paper.

3) The result reported on pp. 12-13 that “DELocal missed 60 differentially expressed TDGs which are identified by all the other methods in [the] RNAseq dataset” is reasonably explained by the authors based on the assumptions of the method. But one can ask if the assumption-defying presence of unrelated differentially expressed genes in the vicinity of TDGs is related to peculiarities of the tooth system. Unlike, for example, cartilage or cardiac muscle, which have (relatively) specific lineage-determining transcription factors (Sox9 and Nkx3-1, respectively), and cell-type-specific products (type II collagen, cardiac myosin HC), tooth development seems to mobilize more generalized regulatory and function-mediating proteins, likely to reside in functionally general chromosome regions. If so, the tooth might represent a stringent challenge that establishes the rigor of the DELocal method which should actually work better for organogenesis involving greater specificity in gene expression.

Minor points

The English usage and some of the phrasing can be improved. There are several cases in which articles are dropped and a few where the meaning is obscure. An example of both:

>Additionally, *the* presence of paralogous genes in the neighbourhood may contradict with *complicate evaluation of* our hypothesis.

Reviewer #2: In this manuscript Das Roy et al., explore whether: 1) The "chromosomal neighborhood" of genes leads to correlated gene expression levels/patterns during organ development; and 2) If so, does accounting for these "chromosomal neighborhoods" improve estimates of (differential) gene expression. I find 1 to be an interesting question and 2 to be important if true. Unfortunately the methods and results do not currently support the conclusions of the authors that both 1 and 2 are true. I see two major concerns that undermine the authors conclusions:

1) The definition of "chromosomal neighborhood" is arbitrary and based on an a priori defined distances. Why should it be that 1Mb is the "right" size of a neighborhood and why should that size be the same for all chromosomal regions? This a priori definition is particularly problematic because there is a biological definition of a neighborhood: Topologically associating domains (TADs) and the smaller compartments (A and B). Previous studies have shown that both of these biologically defined neighborhood vary in size (unlike the 1Mb definition), location (cannot be tiled by 1Mb regions across the genome), and that genes within both TADs and A/B compartments have more or less similar expression levels. Thus, the arbitrary 1Mb regions is problematic. To overcome this limitation the authors should repeat the analyses using regions defined by TADs and A/B compartments from chromosome conformation studies (which might or not might exist for these tissues, but both tend to be conserved across tissues so could be used from other tissues) and/or repeat the analyses using variable distance tiles across the genome.

2) While accounting for 1Mb "chromosomal neighborhood" improves DE results, the effect size is very small from the ROC curves (0.02-0.03) and the data shown in figure 7. But, there is no data presented that these improvements are statistically significant. Please add analyses to demonstrate that these small effect sizes are statistically significant; perhaps bootstrapping the neighborhoods to generate confidence intervals and standard deviations.

The manuscript and its conclusions would be significantly improved should these concerns be addressed and the conclusions still warranted.

**Have the authors made all data and (if applicable) computational code underlying the findings in their manuscript fully available?**

Reviewer #1: Yes

Reviewer #2: **No: **

PLOS authors have the option to publish the peer review history of their article (what does this mean?). If published, this will include your full peer review and any attached files.

Reviewer #1: No

Reviewer #2: **Yes: **Vincent J. Lynch
---

## [Decision Letter · Decision Letter 1]

26 Aug 2021

Dear Das Roy,

We are pleased to inform you that your manuscript 'Chromosomal neighbourhoods allow identification of organ specific changes in gene expression' has been provisionally accepted for publication in PLOS Computational Biology.

Best regards,

Alexandre V. Morozov, Ph.D.

Associate Editor

PLOS Computational Biology

Sushmita Roy

Deputy Editor

PLOS Computational Biology

Reviewer's Responses to Questions

**Comments to the Authors:**

Reviewer #1: The authors have satisfied my main concerns.

**Have the authors made all data and (if applicable) computational code underlying the findings in their manuscript fully available?**

Reviewer #1: Yes

PLOS authors have the option to publish the peer review history of their article (what does this mean?). If published, this will include your full peer review and any attached files.

Reviewer #1: **Yes: **Stuart A. Newman

---

## [Editor Report · Acceptance letter]

3 Sep 2021

PCOMPBIOL-D-21-00670R1 

Chromosomal neighbourhoods allow identification of organ specific changes in gene expression

Dear Dr Das Roy,

I am pleased to inform you that your manuscript has been formally accepted for publication in PLOS Computational Biology. Your manuscript is now with our production department and you will be notified of the publication date in due course.

With kind regards,

Livia Horvath
